# Private Placement of China-Listed Real Estate Firms: A Conceptual Idea

**Yuping Ning *** and **Rohaya Binti Abdul Jalil**

Faculty of Built Environment and Surveying, Universiti Teknologi Malaysia, Johor Bahru 81310, Malaysia; rohaya@utm.my
* Correspondence: ningyuping@graduate.utm.my

**Abstract:** This article conducts a review of the literature on private placement and analyzes the risks facing China's real estate companies. It argues that, within the framework of China's hybrid economic model, private placement can serve as a market-oriented financing mechanism and risk mitigation strategy beyond the traditional banking system. The article focuses on the characteristics of private placement, prevalent hypotheses, and influencing factors. It also traces the evolution of financialization in the global real estate industry, outlines the development model of China's real estate sector, and discusses the challenges and risks it encounters. Private placement offers various advantages, including reducing corporate leverage, strengthening working capital, and addressing information asymmetry issues. However, existing research in this field is still insufficient. Therefore, future research can provide a more robust theoretical foundation and guidance for policymakers, investors, and businesses.

**Keywords:** private placement; risks; real estate; hybrid economic model; real estate; risk mitigation strategy

## 1. Introduction

In the past few decades, the world has been experiencing a wave of real estate financialization, transforming real estate from an illiquid and indivisible asset into one that can be subdivided and traded. Financial institutions, represented by REITs (Real Estate Investment Trusts), have tightly integrated real estate with finance, turning it from a residential necessity into an investable financial asset (Van Loon and Aalbers 2017). The 2008 US subprime crisis exposed the fragility of this system from a sideways perspective (Bernanke 2018; Foster 2008). Although China's process of real estate financialization differs from the global mainstream, a distinct model has emerged under its hybrid economic system, characterized by a mix of national planning and government intervention (Xiong 2023). With land financing playing an essential role, this model revolves around the government, the banking sector, household residents, and real estate enterprises. This combination of real estate and finance has led to remarkable economic success within China over the past four decades. As it gradually becomes a cornerstone of the Chinese economy, this model has also accumulated significant risks. High leverage in real estate companies, an excessive house-price-to-income ratio, and declining birth rates, combined with factors like the US–China trade tensions and the COVID-19 pandemic, have rendered the previous real estate development pattern unsustainable (Fang et al. 2016; Q. Huang 2022a; Rogoff and Yang 2021). Private placement, as a market financing tool, not only reduces the debt burden on real estate firms and enhances corporate governance but also provides working capital for projects and mitigates issues related to information asymmetry.

However, the existing literature fails to definitively answer whether the adoption of private placements as a significant financing method and risk management strategy by real estate enterprises is merely a result of following market trends or a rational strategic

choice. According to Wruck (1989), private placement is defined as a kind of seasonal refinancing mechanism in which companies privately negotiate with and issue equities to a small group of sophisticated investors such as investment funds and banks. In some literature, it can be loosely described as a PIPEs (Private Investment in Public Equity) which is a way for companies to raise capital by selling equity, warrants, or convertibles to private investors. These securities can become publicly tradable after registration, usually within a few months (Chaplinsky and Haushalter 2005). Aside from public offers and rights issues, private placements typically involve a lock-up period that restricts trading, and it aims to issue new shares or bonds as a post-IPO (Initial Public Offering—shares are listed on the exchange for the first time) refinancing option. In this paper, "private placement" specifically means equity private placements.

Since 2006, private placements have increasingly become the primary method of refinancing in China (Wang et al. 2020). According to data from CSMAR (China Stock Market & Accounting Research Database), Chinese listed companies raised a total of CNY 11,011.89 billion (equivalent to around USD 1508 billion) through private placements from 2006 to July 2023. Particularly, real estate enterprises successfully secured CNY 472.16 billion (equivalent to around USD 65 billion), constituting 4.29% of the total financing. Excluding the financial sector, the real estate industry raised more capital than any other sector, following manufacturing, wholesale and retail trade, and transportation, storage, and postal services.

Nevertheless, it is essential to underscore the current deficiency in comprehensive research on the Chinese real estate sector, particularly concerning private placement activities of real estate companies, participant interactions, and the need for a more nuanced analysis of private placement as a risk-mitigation strategy. Our initial incentive is to observe that private placements in the Chinese real estate sector deviate from the findings reported in other scholarly works. For instance, numerous market-level studies in other countries have revealed positive announcement effects coupled with negative long-term stock price performance (Barclay et al. 2007; Hertzel et al. 2002; Wruck 1989). In the real estate Industry, Louisiana et al. (2007) observed a negative announcement effect in the case of private placements of REITs in the United States. Happ and Schiereck (2017) similarly noted a negative announcement effect when examining Seasoned Equity Offerings (SEOs) in 12 European real estate markets (which differ from the US market, as they are not as heavily dominated by REITs), although they did not distinguish between deals executed through private placement. However, X. Zhang (2017) and Tong (2014) reported a more positive announcement effect for Chinese real estate companies. These inconsistencies emphasize the need for a deeper understanding of private placements in the context of Chinese real estate enterprises.

Through a systematic analysis of the characteristics and patterns of global private placement transactions, coupled with an exploration of the distinctive risks inherent to Chinese real estate enterprises, this paper contends that private placement stands as an important strategic tool for Chinese real estate firms at the present stage. It serves as a means to secure refinancing and mitigate risks. From the regulatory perspective, private placement is viewed as a more efficient and market-oriented financing mechanism capable of reducing banks' risk exposure. This marks a significant shift in China's economic development, indicating a substantial transformation in its historical reliance on real estate. In the post-pandemic era, China's real estate sector faces a significant debt crisis, highlighting the importance of studying this issue from this perspective. Discerning and comprehending how participants in real estate firms interact in transactions, leveraging regulatory and authentication roles to mitigate managerial frictions in private equity transactions, constitutes a scholarly issue worthy of attention.

The subsequent sections of this article are structured as follows: Section 2 provides an overview of the regulatory framework of private placement transactions. Sections 3–5 elucidate the private placement phenomenon, prevalent hypotheses, and some important influencing factors, respectively. Sections 6 and 7 delve into the examination of risks within

Chinese real estate companies and the application of private placement as a risk mitigation strategy. Section 8 outlines concerns about private placement, and, lastly, Section 9 furnishes the article's conclusion.

## 2. Private Placement Features

Private placement exhibits distinct characteristics among nations. Table 1 summarizes the states of private placement in some markets. It shows that private placement has grown in popularity in the US, and it is a key form of refinancing tool in other countries, such as Sweden, Australia, India, and New Zealand. Otherwise, placement arrangements are more common among group companies in Japan and Korea, although they are smaller in scale compared to rights issues. (Baek et al. 2006; Kato and Schallheim 1993). Happ and Schiereck (2017) found a similar pattern in European real estate companies. It had relatively small capital, on average and median, compared to other SEOs. However, in China, private placement is in the dominant position among all financing activities (including IPO) and has grown dramatically in the past decades; the scale of fundraising has enlarged tenfold (Shi et al. 2020).

**Table 1.** Private placement around the world.

| Country | Main Findings | Source |
|---|---|---|
| The US | The dollar proceeds increased from USD 8.1 billion to USD 153.9 billion between 1990 and 2000. | Brophy et al. (2004) |
| | The number of cases climbed from 127 to 2719 between 1995 and 2006; overall proceeds rose from USD 1.87 billion to USD 88.0 billion. | Wruck and Wu (2009) |
| Brazil | Between 1995 and 2002, the companies raised approximately BRL 70 million through 653 private placements, in contrast to 123 public offerings that yielded around BRL 25.7 million in capital. | Bordeaux-Rego and Ness (2006) |
| Sweden | Private placement accounted for about half of the SEOs. | Cronqvist and Nilsson (2004) |
| The UK | Private placements had overtaken rights issues as the most important means of refinancing for British companies. | Armitage and Snell (2001) |
| Australia | The capital raised increased 20-fold from AUD 2.3 billion to AUD 46 billion from 1995 to 2009. | Xu et al. (2017) |
| New Zealand | The average private placement proceeds were NZD 779.3 million with an average proportion of outstanding shares of 8.4%. | Anderson et al. (2006) |
| India | The average size of a private placement was about USD 75.94 million. | Katti et al. (2020) |
| East Asian (Japan and Korea) | Private placement arrangements were more common among group companies, and they accounted for a smaller share of new equity issuance compared to rights issues. | Kato and Schallheim (1993); Baek et al. (2006) |
| China | In 2013, an estimated 263 listed companies employed private placement, with total proceeds of CNY 300 billion (USD 46.133 billion). | Tao et al. (2018) |
| | There had been an increasing number of publicly traded firms offering private equity placement (from 52 in 2006 to 505 in 2017), and the scale of fundraising had enlarged tenfold during this time. | Shi et al. (2020) |

### 2.1. Regulations

Companies' financing behavior is guided by two important rules in the US. The "Securities Act 1933" specifies the rules for declaring and reporting securities for a firm that sells its stocks to the public at a fixed price. The "Securities Exchange Act 1934" emphasizes the need for a company's periodic disclosure. The United States Securities and Exchange Commission (SEC) is not required to register shares that are ordinarily issued through a private placement. However, reselling those shares after the one-year lock-up period necessitates SEC registration. In 1990, "Rule 144A" was introduced by the SEC, which allowed

the selling of private placements to "Qualified Institutional Buyers" (QIBs) without the securities being registered or held for a year, as previously required (Eckbo 2007; SEC 1971).

Regulatory regimes are similar across countries, and laws and regulations are generally drafted regarding US practices. For instance, the Japanese Commercial Code established criteria for new stock offerings, requiring the issuing business to disclose the class, quantity, and issue price of the new shares to the public two weeks before the payment date (Kato and Schallheim 1993). In Sweden, there is little legal or institutional difference between a private placement and a rights offering (Cronqvist and Nilsson 2004). Australia's regulations are relatively lenient, with a cap of 15% on fundraising and no restrictions on the scale of discounts or resale (Xu et al. 2017), a framework similar to that of New Zealand. Additionally, New Zealand's existing disclosure regulations permit buyers to engage in transactions within a specified timeframe without the obligation to publicly announce their activities to the market (Anderson et al. 2006). Under the SGX's (Singapore Exchange Limited) continuous listing guidelines, only rights offerings and private placements are to be made in Singapore, where public offerings are not permitted (Tan et al. 2002). Katti et al. (2020) stated that, in India, shares are exclusively sold to qualified institutional investors (QII) and promoters are not permitted to participate in the sale. The offer's floor price is determined by the regulator, and QII investment in private placements is not subject to a lock-in term in India.

### 2.2. The Rise of Private Placement in China

Private placement and public offering are the two most important forms of equity refinancing. In 2013, SEOs accounted for 86% of equity financing, far exceeding IPO and rights offerings (Hu and Zhang 2016). Since the emergence of private placement in 2006, it has become the dominant financing method (J. Wu 2016; Tao et al. 2018; Zhang et al. 2021).

Regulations of China

On 20 February 2023, the CSRC (China Securities Regulatory Commission) fully implemented the registration-based system. However, previous studies mainly explored the EA System (An Examination and Approval System), where refinancing decisions must be approved by the CSRC. The implementation of a registration-based system holds various positive implications. It signifies a restructuring of regulatory authority within CSRC, involving the delegation of substantial review powers to local government entities, intermediaries, and other stakeholders, thereby enhancing societal efficiency (Jiang 2014). The registration-based system places a pronounced emphasis on information disclosure, thereby standardizing the conduct of publicly listed companies, and actively contributing to investor protection and the maintenance of order in the securities market.

Figure 1 demonstrates the timeline of private placement in China. Before making the private placement plan public, the company will conduct a feasibility study to discuss the refinancing plan as well as the risk assessment. The terms and conditions of the proposal will be presented to the shareholder meeting for approval and then the firm will submit the proposal to the CSRC for approval. Following CSRC clearance, the business will collect the capital from the purchasers, and the formal announcement with details will be made immediately after the money is obtained and the newly issued shares will be registered. According to this timeline, several important dates will be clarified. First is "Announcement Day", which is the date that the proposal is publicly announced. Secondly, the "Firm Board Approved Date" is the date the proposal is approved in the shareholders' meeting. The third day is the "Approval Date" when the CSRC approves the private placement proposal handed by firms. The fourth is "Implementation Day" when the private placement plan is implemented. The last one is "Traded Day" when the newly issued shares end their lock-up period and can be freely traded in the exchanges.

On announcement day, Chinese enterprises normally specify the lower bound of the issue price disclosed after the agreement is completed. To deal with such possible unfairness, the authorities, on the other hand, established a more rigorous guideline for

pricing. Generally, the lower bound is set as 90% of the average price 20 trading days before the settlement date, which is termed the "90% Rule". On 17 September 2007, the CSRC published a new manual, which was set to calculate the issuing price.[1] Authorities aimed to replace the 90% rule and regulate how the firms conduct their private placement deal. It further required that there be no more than 10 buyers, which could be mutual funds, trustees, foreign strategic investors, individual investors, and other legal investment organizations. The newly issued shares are locked up for 1 to 3 years, depending on the investor type.[2]

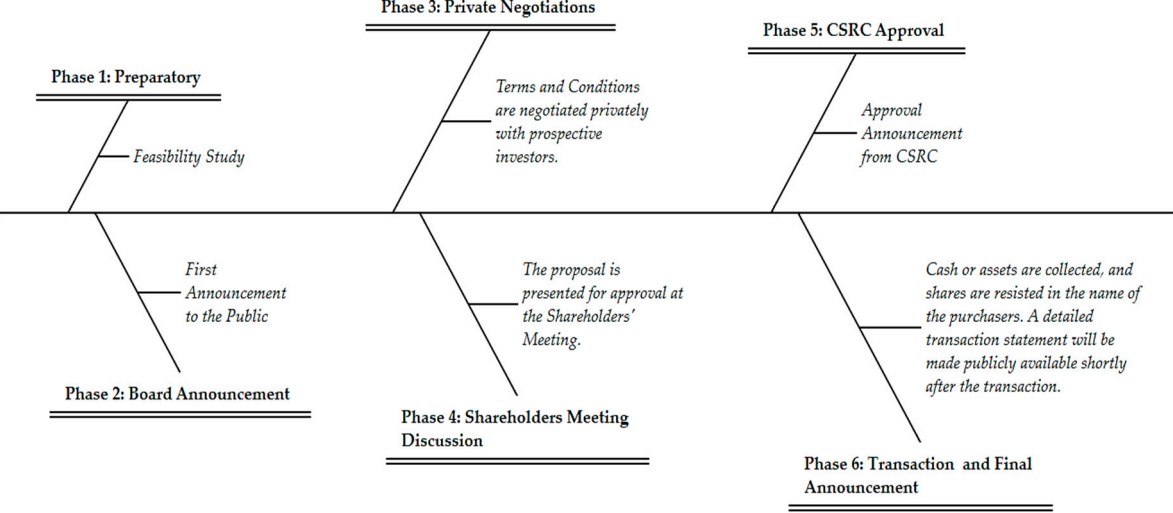

**Figure 1.** Timeline of Private Placement in China (Source: Song (2014)).

In October 2015, the CSRC's Issuance Department issued the latest "Window Guidance Opinions", encouraging the issuing firm to employ "Implementation Day" as their basis for determining the issue price. The authority indicated that such transactions could be directly submitted to the preliminary review meeting if there were no major problems. In February 2017, the CSRC announced a new regulation named "The Amendment on The Issuing Measures of Non-Public Shares of Listed Companies", which formally confirms "Implementation Day" as the benchmark day. On 14 February 2020, CSRC issued three new amendments.[3] This primarily entailed changing the threshold of 90% to 80%, in which the lower bound became 80% of the average price 20 trading days before the settlement date. In the previous version, there were no more than ten buyers on the main board and no more than five purchasers for GEM (Growth Enterprise Market) listed businesses, whereas today there are no more than thirty-five buyers for either. If a listed business asks for a non-public offering of shares, the number of shares to be issued shall not exceed 30% of the entire share capital before the current issue, compared with 20% prior.

Compared to other SEOs, private placement in China has more merits. Normally, the other SEOs have restricted requirements on issuance. Companies must meet the requirements of profitability, financial status, and the dividend policy when implementing other SEOs, while there are no similar requirements for private placement (Xu 2011; Yu and Luo 2015). Private placement does not need to go through the procedures of publishing prospectuses and public inquiry, so it saves a great time and procedure to operate (W. Zhang 2008). The company may use private placement for strategic goals, including asset injection for industrial integration, raising funds for acquisitions, issuing additional equities for high-quality assets to facilitate parent company listing, introducing strategic investors for enhanced external governance, and minimizing stock price impact due to lock-in periods (Yu et al. 2016; Zhang and Guo 2009; Xu 2011; Yi et al. 2006; Zhou 2016). Compared with other refinancing methods, private placement has the advantages of low issuance cost (Xu 2011), flexible pricing mechanism (Tao et al. 2018; W. Zhang 2008), diversified subscription methods (Xu 2011), the premium of the parent company's assets value through private

placement implemented by a subsidiary (Yi et al. 2006), and ease of approval (W. Zhang 2008; Zhou 2016).

### 3. Private Placement Phenomena

The well-documented phenomena of private placement are the significant announcement effect (Wruck 1989), the downturn of long-term stock performance (Hertzel et al. 2002), and substantial discounts (Hertzel and Smith 1993).

#### 3.1. Announcement Effect

The most idiosyncratic characteristic of SEOs is the negative announcement effect in various nations and industries (e.g., Veld et al. (2020)[4] for various markets; Eckbo and Masulis (1992) in the US; Hansen (1988) in the right issue; Happ and Schiereck (2017) in the European real estate industry; and Tan et al. (2002) in Singapore) because the market interpretation the SEOs as a signal that the stock price is overvalued (Myers and Majluf 1984).

In contrast to SEOs, private placements often yield a positive abnormal return. In the US, private equity sales show a 4.5% average abnormal return (Wruck 1989). Hertzel and Smith (1993) purported a 1.7% average price run-up during [−1, 0], increasing to 5% when participants are actively involved in future operations. Barclay et al. (2007) reported a larger positive announcement effect with public interaction, such as joint ventures. Consistent findings are seen in works by Marciukaityte et al. (2005) and Hertzel et al. (2002). In Japan, Kato and Schallheim (1993) stated a significantly positive announcement effect of around 5%, while, in Singapore, a 21-day average abnormal return of approximately 6% is documented (Tan et al. 2002). Cronqvist and Nilsson (2004) discovered positive (7.3%) and significant reactions to private placements in Sweden. In China, W. Zhang (2008) reported statistically significant average CARs of 11.87%, 8.592%, and 7.199% in the windows of [−20, +5], [−10, +5], and [−5, +5], respectively, consistent with other Chinese scholars (Sun 2015; J. Wu 2016; Wei and Na 2008; Hu and Zhang 2016; Liu 2008; Zhang and Guo 2009).

#### 3.2. Long-Term Performance

Private placements are well proven to have negative long-term performance. For example, Hertzel et al. (2002) found industry-adjusted profitability at −0.107 for year 1 and −0.086 for year 2. Barclay et al. (2007) reported a significant negative price rundown of −9.8% in the time interval [−1, 120], which they attributed to passive investors' increased entrenchment. Wruck and Wu (2009) reported the average three-year match-adjusted return is significantly negative at −25.27% ($p = 0.00$) which is similar to the findings of Krishnamurthy et al. (2005), who report −38.39% for a similar interval. In Asian markets, from 1989 to 2000, Korean businesses saw an average −42.34% cumulative abnormal return in the event window of [−10, 480] (Baek et al. 2006). Kato and Schallheim (1993) observed a significant negative price movement of 7.53% in Japan within the specified period [16, 100]. This finding aligns with the results of an earlier study by Kang et al. (1999). In China, private placement normally has an undesirable influence on long-term stock performance (Geng et al. 2011). Yu et al. (2016) claimed that, although the short-term effect is favorable, the medium-term windows [0, 180] and [0, 360] had cumulative returns of −2.6% and −4.6%, respectively, and the longer-term holding periods [0, 540] to [0, 900] had cumulative returns of −5.8% to −11.8%.

#### 3.3. Discount

For SEOs, Rock (1986) proposed that firms sell shares at a discount to informationally disadvantaged investors.[5] Similarly, it is widely reported that the equity sold at a substantial discount in a private placement. SEC (1971) reported average discounts of about 30% for unregistered shares in the US. Hertzel and Smith (1993) found an average of 20.14% in the samples, which is in line with others (Barclay et al. 2007; Chu et al. 2005; Silber 1991; Wruck and Wu 2009; Wruck 1989) and Lim et al. (2021) in PIPEs. Floros and Sapp (2012) observed a declining trend in discounts, which declined from 24.76% in 1995 to 5% in 2008,

which is consistent with Huson et al. (2009), in which PIPE discounts decreased from an average of 16.4% (1995–2000) to 9.8% (2001–2007). Substantial discounts are also observed in other markets (Xu et al. (2017) found 7% in Australia; Kato and Schallheim (1993) discovered 11% in Japan). In China, Tao et al. (2018) reported the average discount was 32.88% on managerial placements and 20.784% on non-managerial placements, whereas Zhang and Guo (2009) ascertained the block is even sold at a discount of 60.7% when all new equity is sold to the controlling shareholder. On the other hand, there are few markets sold at a premium. Tan et al. (2002) corroborated a significant 13.73% premium for a private placement in Singapore, whereas an average 4.37% premium was reported in India (Katti et al. 2020). In New Zealand, the pricing of placement is mixed, in which 27% of the sample is placed at an average premium of 4.7%, in contrast to 73% of the sample that is placed at an average discount of 10.2% (Anderson et al. 2006). Furthermore, there is a relatively small number of property companies issuing equity privately at a premium in Europe (Chu et al. 2005).

## 4. Theoretic Framework

The actions of management, controlling shareholders, and other investors are important pieces of evidence to look at since they may be interpreted from the alignment and the expropriations standpoint, especially for management and controlling shareholders. Table 2 demonstrates some of the leading hypotheses about private placements.

### 4.1. Agency Theory

In the context of agency theory, Jensen and Meckling (1976) emphasized the separation of ownership and control within contemporary firms, which can lead to conflicts where management may prioritize their interests over shareholders. The alignment of interests between management and shareholders is crucial for efficient resource allocation and firm value improvement. The monitoring hypothesis suggests that a positive stock price reaction to announcements is indicative of enhanced supervision by external investors, and discounts serve as compensation for this monitoring service (Wruck 1989). Shleifer and Vishny (1986) shared a similar perspective, attributing increased firm value to effective management oversight or professional advice. Morck et al. (1986) highlighted that the concentration of ownership can have varying effects on firm value, with larger shareholders having a greater role in monitoring management. The entrenchment hypothesis, proposed by Barclay et al. (2007), posits that managers can use their authority to sell equity for entrenchment, which can lead to negative market responses. The term "tunnelling" is introduced to describe the expropriation of minority shareholders' assets and is more prevalent in civil law countries, particularly in Asian nations (La Porta et al. 1999, 2000). Some scholars argued discounts in China's private placement are employed for tunnelling or expropriating minority shareholders' and external financial providers' benefits (Zhu et al. 2008; Zhang and Guo 2009; Xu and Xu 2011).

### 4.2. Certification Hypothesis

Myers and Majluf (1984) suggested that management may strategically time the market to issue shares at a higher price when the company is overvalued, resulting in negative announcement effects for SEOs. Hertzel and Smith (1993) argued that shifts in market perceptions of a company's value and potential investment opportunities play a role in the valuation changes. Private placements are seen as a means to address information asymmetry between the firm and the market, with the participation of experienced and sophisticated investors serving as a certification that the firm is undervalued. Managers may employ private placements to convey such messages to the market by avoiding higher information costs and risks of information leakage involved in public offerings.

**Table 2.** Hypotheses of private placement.

| Hypothesis | | Main Finding | Authors |
|---|---|---|---|
| Agency Theory | Monitoring Hypothesis | Positive stock price reaction to announcements reflects enhanced supervision by external investors; discounts in private placement can be seen as compensation for this monitoring service. | Wruck (1989) |
| | | Large shareholders play a key role in monitoring management performance when ownership concentration increases. | Morck et al. (1986) |
| | Entrenchment Hypothesis | Managers have authority over when and to whom equity is sold, potentially for entrenchment purposes. The market may respond negatively when the market understands the managers' intent. | Barclay et al. (2007) |
| | | Managerial placement is consistent with the entrenchment hypothesis but not the certification hypothesis. | Tao et al. (2018) |
| | Tunnelling Hypothesis | Major shareholders in China seize the interests of minority shareholders through private placement. | Zhang and Guo (2009) |
| | | Discounts in China's private placement are employed for tunnelling or expropriating minority shareholders' and external financial providers' benefits. | Zhu et al. (2008); Xu and Xu (2011) |
| Certification Hypothesis | | Private placement helps address information asymmetry between the firm and the market. Participation by experienced and sophisticated investors is seen as a certification that the firm is undervalued. | Hertzel and Smith (1993) |
| | | The market perceives the issuance of PIPEs as a positive certification signal. | Floros and Sapp (2012) |
| Over-optimism Hypothesis | | Investors tend to get overly optimistic when events occur; investors may base their predictions on a company's future success on comparable previous success stories. | Loughran and Ritter (1997) |
| | | When the market is more optimistic, the announcement effect of private placement is more noticeable. | Marciukaityte et al. (2005) |
| | | Investor sentiment significantly impacts the discount of private placement; discounts may be even greater in the presence of over-optimism. | Lu and Li (2011) |
| | | The impact of investor sentiment on private placement discounts is particularly pronounced in situations of intense market sentiment, such as bull and bear markets. | Li and Jian (2017); Wang et al. (2021); Yu et al. (2016) |

*4.3. Over-Optimism Hypothesis*

Drawing from behavioral finance, De Long et al. (1990) proposed models suggesting that erroneous perceptions generated by noise traders can lead asset prices further away from their underlying value, deviating from the predictions of earlier theories. Hertzel et al. (2002) contended that investor over-optimism about a firm's future can result in unfavorable market reactions, including declining prices and poor operating results. This over-optimism may stem from a tendency among investors to become overly optimistic when events occur, giving precedence to current events over future expectations (Marciukaityte et al. 2005). Ritter and Loughran (1995) highlighted that investor over-optimism often leads to a focus on recent success and a tendency to prioritize present events over future outcomes. Investor sentiment has a significant impact on private placement discounts, with greater over-optimism potentially leading to larger discounts. These discounts compensate for overvaluation and contribute to long-term return mean reversion, particularly in situations of intense market sentiment.

*4.4. Argument on Hypotheses*

Figure 2 depicts the explanations via the lenses of agent theory, behavior finance, and asymmetric information. According to agency and asymmetric information theories, the private placement's favorable announcement effect is accounted for by participants' provision of monitoring and certification service. It also exhibits exploration, entrenchment, and tunnelling effects, reflecting the tensions and frictions among the stakeholders. Because of its low communication cost, private placement is suitable from the standpoint of information asymmetry theory for enterprises with substantial information asymmetry difficulties to get valuable financing. The monitoring and certification impact may be seen when these insiders make judgments from an alignment standpoint. Both negative and long-term, post-announcement operating and stock action are widely documented (Hertzel et al. 2002), which is hardly explained by agency theory and certificate hypotheses. Participants' motivations can be indirectly deduced from market responses, assuming that markets are rational and trade reactions are accurate. However, it is debatable if less mature markets, like those in East Asia, are truly efficient, questioning the validity of this assumption. Barclay et al. (2007) argued that both the monitoring and certification hypotheses suggest that bulk buyers offer important monitoring or certification services at discounted rates. They found it puzzling that such significant price discounts adequately reflect continuous monitoring services or a one-time certification. The long-term underperformance of stocks challenges the notion that these companies are of high quality, but some attribute this to inefficient markets, suggesting that the market misinterprets signals.

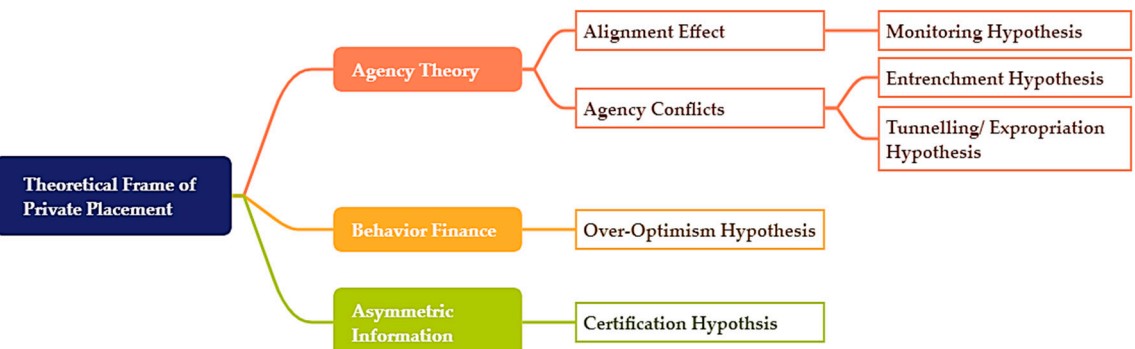

**Figure 2.** Summarize of private placement hypotheses (source: author).

The tunnelling and entrenchment hypotheses struggle to explain the observed disparities. This may be because the alignment effect dominates when monitoring is key in the relationship, whereas agency frictions lead to entrenchment or tunnelling effects. However, it is illogical to assume certain effects will dominate in the pre-set stage. For instance, the alignment effect typically appears during the event, with negative effects surfacing post-event. From the standpoint of behavioral finance, the over-optimism hypothesis is the explanation that makes sense logically on its own. Irrational investor optimism can be used to explain short-term stock returns, while value regression can be used to explain long-term stock returns. they claim the discount as a way to make up for low long-term returns. However, the phenomenon of the consistency between the long- and short-term price performance in some markets does exist (Anderson et al. 2006). Naturally, such excessive emotional reactions in the market should be universal, and there will be no significant difference among the markets. Hence, the idea that the market controlled its "emotional" response is implausible. From this perspective, more efforts should be put into enriching the theoretical explanations.

## 5. Influence Variables

Private placement, as a refinancing method and risk-mitigation strategy for real estate enterprises, is influenced by several key factors. These factors include cash-flow liquidity,

financial distress, leverage, concentration of ownership, information asymmetry, and the roles played by major shareholders, as well as the nature of ownership.

The decision to refinance is partly the result of companies in financial distress having to find money to bail them out. As a financial resource, private equity is an important financial resource in addition to the banking system and internal financing (Cronqvist and Nilsson 2004; Floros and Sapp 2012). Additionally, the company's cash flow may worsen as a result of the public disclosure of the company's bad operations, which could prompt demands for advance payments from banks and suppliers as well as worries from customers about the company's insolvency and lack of follow-up, after-sales support. To stop these disasters, private placement might be used to prevent the disclosure of this internal information.

Due to the rigid constraints of debt, highly leveraged businesses will put more pressure on management; in turn, management would prefer to issue shares to reduce the burden. Since bonds are typically used for less leveraged companies, high-leverage firms struggle to collect debts from the conventional method and, therefore, are willing to pay a significant discount to issue shares. Zhang et al. (2021) stated that the discount is negatively influenced by the leverage in Chinese listed companies, which implies the substantial discount may represent the risk involved with the issuing companies. Zhang et al. (2021) and Sun (2015) pointed out that there is a negative association between the leverage and discount price in the SEOs of China-listed businesses.

Smaller firms are more likely to have larger discount and discount-adjusted abnormal returns than larger ones because of greater information asymmetry (Hertzel and Smith 1993). Since information production is subject to economies of scale, the discounts should be larger for those that have a substantial information asymmetry problem, such as smaller companies, start-ups, and businesses in the early stages of development (Gomes and Phillips 2004; Hertzel and Smith 1993; Szewczyk and Varma 1991; Y. Wu 2004). Katti et al. (2020) found larger issues are positively associated with firm size, which has a lower level of information asymmetry. The low negotiation costs of the private placement, combined with management's decision to avoid a public offering, indicate the market management's perception that the company is undervalued (Happ and Schiereck 2017). Some scholars believe the high levels of asymmetry information are the main reason to choose private placements rather than SEOs (Chemmanur and Fulghieri 1999; Chen et al. 2010; Gomes and Phillips 2004; Schultz and Twite 2016; D. M. Wu 1974).

Since private placement will inevitably change the distribution of ownership, it is natural to prioritize the ownership concentration brought about by the placement and its impact on the valuation of the company. Wruck (1989) corroborated that the announcement effect is correlated with ownership concentration when it is less than 5% and larger than 25%. She revealed that changes in the concentration of ownership had a positive impact on the announcement effect of a private placement. Hertzel and Smith (1993) argued that the enhanced concentration of ownership may occur with the firms which have a higher level of concentration initially.

Major shareholders in private placement may play a more active role. Wruck and Wu (2009) found that the market responds more positively to placements with dominant investors. The ownership of the largest controlling shareholder has a positive impact on the performance of the enterprise in China (J. Wu 2016; Zhang and Li 2008; Chen and Xu 2001). Wang et al. (2020) reported a strong link between different pricing mechanisms and discounting for controlling shareholder participants. Yang and Sun (2017) confirmed the participation of controlling shareholders has a higher stock reaction to a private placement in China, yet the finding is inconsistent with Hu and Zhang (2016). Further, the participation of controlling shareholders conveys the signal that the block-holders are more optimal for the future of the enterprise, and the market will think the participation can effectively reduce the asymmetric information problem as well (Brown and Floros 2012; Chen et al. 2010; Dai 2007; Yu et al. 2016).

The effects of ownership differences on private placement are not well covered in the literature. Some literature, however, asserts that the announcement effect is positive regardless of the type of ownership, but state-owned enterprise performance is inferior to that of private ones in both the long and short term. (Huang and Wu 2015; X. Huang 2017; Peng 2013). Huang and Wu (2015) believed one possible explanation for this is that state-owned businesses have greater social obligations overall. They reported that state-owned enterprises are much more than private ones in terms of the number of cases, shares issued, and the actual amount of funds raised. Peng (2013) found that state-owned enterprises' short-term price and market performance were weaker than those of private enterprises when adopting the private placement of asset injection. Shleifer and Vishny (1994) ascertained that the relationship between businesses and politics could affect enterprise value. Johnson and Mitton (2003) discovered that Malaysian businesses with political ties are more likely to be approved for bank loans. Consequently, it makes sense to assume that businesses with specific political connections might find it easier to get approved for private placements. This idea is attested by Yang et al. (2016). They found that these unlisted companies had a higher private placement success rate than unaffiliated companies.

For other influencing variables, Wruck and Wu (2009) noted a positive correlation between dollar proceeds and discounts, contrasting the findings of Hertzel and Smith (1993) on discount prices and firm size. Zhang and Li (2008) emphasized the dual objectives of Chinese listed firms' private placements for overall group listing and addressing related transactions. Xu and Xu (2011) found that private placement for mergers and acquisitions effectively minimizes financial strain and transaction costs. Silber (1991) argued that the discount compensates for illiquidity, contrasting with Barclay et al. (1993)'s view on excessiveness. Cheng et al. (2014) discovered the outperformance of long-term institutional investors in Taiwan. Brophy et al. (2009) noted hedge funds' importance, especially in financing companies with poor fundamentals, since they protect themselves mainly through a wealth of hedging tools, deep discounts, negotiating repricing power, and shorting underlying stocks. Floros and Sapp (2012) proposed that board-involved investors actively supervise firms in PIPEs. Anglin et al. (2011) found that enhancing financial incentives for board members minimizes knowledge asymmetry. In Indian firms, CEO duality drives the private placement premium, while board size has no impact (Katti et al. 2020). Li et al. (2011) observed an adverse relationship between managers' age and tenure and investment scale in state-owned firms, contrasting with the finding of Jiang and Dai (2009), where there is a positive correlation between manager age and corporate performance.

## 6. Risks in China's Real Estate Sector

In the past few decades, the global economy has witnessed a wave of financialization, digitization, and securitization. With economic growth, households in developed countries such as Europe and the United States have accumulated substantial wealth. They have been increasingly entrusting these funds to more specialized financial management funds and pension funds in search of better returns. Although real estate inherently possesses characteristics of illiquidity, indivisibility, and long investment horizons, financial securitization has transformed real estate into a tradable and highly liquid financial asset (Van Loon and Aalbers 2017). However, the transformation of real estate into a wealth and investment pursuit, detached from its residential essence, poses significant risks not only to individuals but also to the stability of the financial system. For instance, Haila (2021) explained how securitization turns real estate mortgages into saleable securities, injecting liquidity into properties that are inherently non-divisible and illiquid. REITs further break down ownership of managed real estate into smaller securities sold to other investors and banks. Financial entities of this nature can become more dynamic when homeowners fail to repay their loans and lose their homes. In addition, electronic trading and speed trading can be used to enhance the profitability of real estate financial products. Blakeley (2021) argued that real estate financialization results in broader wealth instability and homelessness.

Loose monetary policies post-financial crisis have driven up housing prices, accelerating the accumulation of potential financial risks, especially in the context of the COVID-19 pandemic. Wijburg et al. (2018) described an interesting scenario in the rental market. In a complex system constructed with various financial actors and a dizzying array of real estate financialization products, "...which makes it difficult to conceptualise who really is the landlord and to whom tenants should address their grievances." Furthermore, the financial crisis triggered by the real estate bubble in 2008 was largely attributed to the accumulation of risk resulting from housing mortgage securitization (Foster 2008). Despite Bernanke (2018) asserting that the severity of the 2008 Great Recession was primarily due to panic in the financing and securitization markets, it indirectly underscored the significant costs paid by the complex interplay of real estate and finance to maintain stability, much like a sandcastle constructed by children at the beach, the taller it is built, the more precarious it becomes.

Aveline-Dubach (2020) compared the differences in real estate financialization between Japan, Hong Kong, and mainland China. He suggested that the Chinese government has deliberately steered away from the global mainstream financialization process, choosing instead to allow the development of localized real estate financialization. Xiong (2023) argued that China's unique hybrid economic structure, combining market mechanisms with state planning and government intervention, makes it less likely to experience a financial crisis similar to that of the United States in 2008. He explained that over the past four decades, China's economy had evolved into a hybrid of private and state-owned enterprises, and through several rounds of reforms, state-owned enterprises have become more profitable and dominant in certain strategic industries. On the other hand, government intervention is also aimed at enabling private enterprises to harness their vitality. For instance, in September 2023, the government established the "Private Economic Development Bureau" to assist private enterprises. Through regular communication with businesses, it aims to help them address operational challenges and enhance their international competitiveness. Simultaneously, a series of policies aimed at revitalizing private enterprises and attracting foreign investment have been introduced, showcasing the government's steadfast commitment to the path of reform and opening up. These measures are designed to guide investor expectations through policy directives.

When this hybrid economic model operates optimally, with a balance between state intervention and the market, it can mitigate market externalities and enhance economic efficiency. Xiong (2023) illustrated the real estate industry in the hybrid economy. In simple terms, local government officials carry out orders from the central government while aiming to promote local economic development for career advancement. They actively develop the local economy by selling land at high prices to raise capital for repayment through loans from financing platforms (LGFVs, Local Government Financial Vehicles). Real estate, with its land and unsold properties, is used as collateral for bank project financing and financed using homebuyer loans. Household sectors use their future income to repay loans from banks. As for the role of government-led public investments, it can directly or indirectly create favorable conditions for private investments (Xu and Yan 2014). For example, improving private investment productivity through infrastructure development provided by public investment reduces production costs, positively impacting the profitability of private investments. Local governments can enhance the attractiveness of local communities by building schools, expressways, hospitals, parks, and other infrastructure, contributing to higher local housing prices and, consequently, increased land prices, allowing them to obtain more capital for improving social welfare and promoting urban attractiveness. Moreover, the massive upstream and downstream business chains associated with the real estate industry have a significant impact on local employment and economic development. Motivated by successful experiences in other regions and the pressure for career advancement within the political systems, local officials have vigorously developed "Land Finance" in recent decades, achieving remarkable results.

However, this hybrid economic model under state planning and government intervention has brought about certain issues. High housing prices have squeezed out household consumption in other areas. For instance, Fang et al. (2016) argued that purchasing a house requires savings equivalent to 3.2 times a household's annual income for down payments and an additional 45% of annual income for mortgage repayment. Even in second and third-tier cities, the housing-price-to-income ratio has reached alarming levels, with Beijing, Shanghai, and Shenzhen occupying three of the top four positions globally in 2018 (all exceeding 40, while New York had approximately 12), with Hong Kong taking the third position (Rogoff and Yang 2021). If income growth continues, the cumulative household leverage will gradually decrease over the long term. Household incomes have been growing rapidly over the past two decades (Fang et al. 2016); however, the slowing economic growth resulting from the US–China trade war and the COVID-19 pandemic has altered household expectations for future income, upsetting the balance of the real estate and financial system under the hybrid model. The conclusion of China's urbanization process and the premature onset of population ageing have reduced the demand for real estate. High inventories, high housing prices, high new housing construction areas, and high per capita residential areas combined with the marginal effects of dilapidated housing redevelopment have gradually weakened, making the real estate crisis fundamentally different from the previous situation (Fang et al. 2016; Q. Huang 2022a, Rogoff and Yang 2021; Xiong 2023).

Another significant risk in the Chinese real estate industry is the high exposure of banks and the high leverage of real estate enterprises. One-quarter of the debt in the banking system is related to real estate, with half of it connected to local governments (Xiong 2023). In Figure 3a, funds obtained by real estate enterprises through domestic loans have steadily increased since 2006, reaching a peak in 2020 when a total of CNY 2667 billion was acquired. In contrast, the funds obtained through private placements were comparatively less, peaking in 2015 at no more than CNY 180 billion. Figure 3a indicates that the mainstream source of funding for domestic real estate comes from the traditional banking system, with significant potential for increased funding through capital markets. Panel b reveals two peaks in the declaration of private placement transactions by real estate enterprises: one during the period from 2006 to 2009 and another around 2015. After the pandemic, various measures were introduced to encourage the development of the real estate industry to prevent economic sluggishness. For example, in August 2023, CSRC suspended the approval of private placement, yet refrained from imposing restrictions on refinancing activities for real estate enterprises. As of November 2023, a total of 20 real estate enterprises have declared their intention to raise funds through private placements in the capital market. It seems there is a renewed enthusiasm among real estate enterprises for implementing private placements.

Figure 3c illustrates the continuously increasing corporate leverage since 2008 reaching more than 80% in 2020. The high leverage of real estate has raised concerns at the central government level about financial risks, leading to the introduction of the "Housing Is For Living, Not For Speculation" policy in December 2016 and the "Three Red Lines" policy in August 2020. However, these policies, combined with the impact of the pandemic, have exacerbated the operational difficulties of real estate companies, ultimately leading to a real estate debt crisis, exemplified by Evergrande (D. Wu 2022; F. Zhang 2021; Liu et al. 2022; Feng and Ge 2021; Wang 2021; Zhao et al. 2017). With other economic problems stemming from the real estate industry, policymakers are contemplating alternative development paths, seeking to transition from the current, real estate-based economic structure to a healthier and sustainable development path (Q. Huang 2022b). Continuing to provide real estate financing through traditional banking systems not only exacerbates the banks' risk exposure but also increases the debt burden of real estate enterprises. It also keeps local governments tied to the constraints of land finance. Property taxes are, of course, a potential solution, but Xiong (2023) argued that the collection of property taxes may face opposition from residents and local governments. For instance, to adjust the expectations

of the market, the central government announced the postponement of legislation on real estate taxes in September 2023. This complex interplay between real estate and finance in the hybrid economic model with state planning and government intervention has raised several issues, including high housing prices, excessive bank exposure, and real estate company leverage. Policymakers are now exploring alternative development paths to transition from the current real estate-driven economic structure to a more sustainable and balanced one.

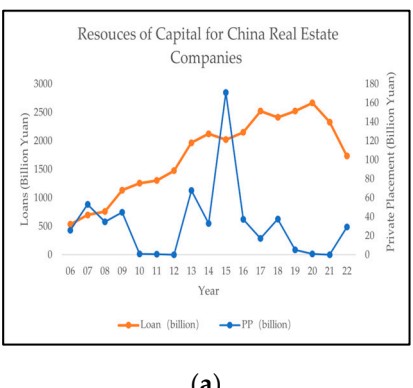

(**a**)

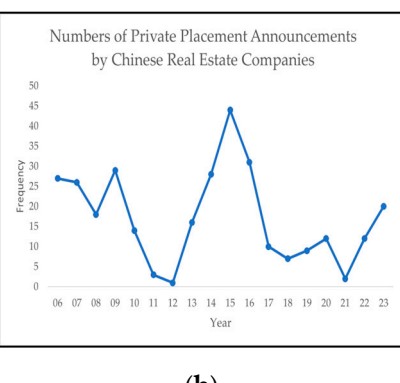

(**b**)

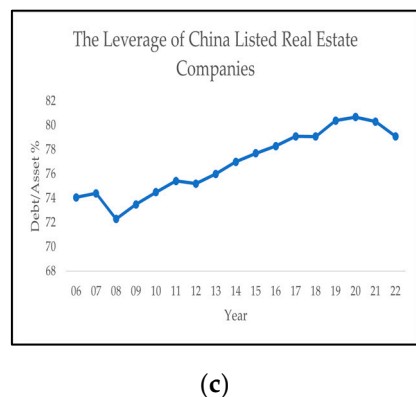

(**c**)

**Figure 3.** The statistics of China-listed real estate companies: (**a**) A comparison between the funds obtained by real estate enterprises through domestic loans and private equity placement. Source: CSMAR database; (**b**) The annual numbers of private equity placements declared by listed Chinese real estate companies.[6] Source: Tonghuashun Wencai database; (**c**) The debt/asset ratio from 2006 to 2022 for China real estate companies. Source: CSMAR database.

## 7. Private Placement as a Risk Mitigation Strategy

Private placements, as a means to help real estate companies, reduce leverage, and obtain valuable working capital, play a significant role. From a macro perspective, private placements represent a more market-oriented approach. Since the issuance targets a small number of sophisticated investors with specific knowledge backgrounds, the overall societal cost of investigating the true value of companies and projects is inevitably lower than public offerings. Moreover, the expertise and financial strength of the parties involved in the transaction give them an advantage in negotiations over ordinary investors. This advantage allows both parties to engage in a negotiation over the transaction price and the total funds raised, reducing the problem of overinvestment. From the government's perspective, expanding alternative financing channels beyond traditional bank financing is a means of resource allocation through market mechanisms. By doing so, the government can obtain market information in a more timely manner and at lower costs through the signals sent by the market. Additionally, this approach prevents the worsening of banks' exposure to the risks of the real estate sector and the leverage ratios of enterprises.

From a micro perspective, private placements can impact real estate companies in terms of cash flow, leverage, and investment opportunities. Figure 4 illustrates the relationships among these aspects, namely, having sufficient and necessary cash flow, obtaining essential financing for new projects, and gradually reducing the company's debt ratio. It is the key to finding suitable investment opportunities with sufficient returns as China has undergone an economic slowdown in recent years. Widespread agency conflicts within companies often lead to involvement in ineffective investment projects or the imposition of substantial costs. The information asymmetry between the industry and capital can be mitigated since capital with a keen investment awareness can help real estate companies find investment opportunities through its extensive network. Traditionally, the cash flow in the real estate industry experiences tight constraints. The recent debt crisis has exacerbated public concerns about the depletion in cash flow in real estate companies. However, debt financing would further increase the leverage ratio of enterprises, compelling them to en-

gage in more irresponsible project investments. The operating cash flow of companies will then deteriorate in a spiraling manner. According to the pecking order theory of financing, when debt financing is not feasible, equity financing must be considered. Compared to other sources of equity financing, private placements offer significant advantages and are likely to be a viable solution.

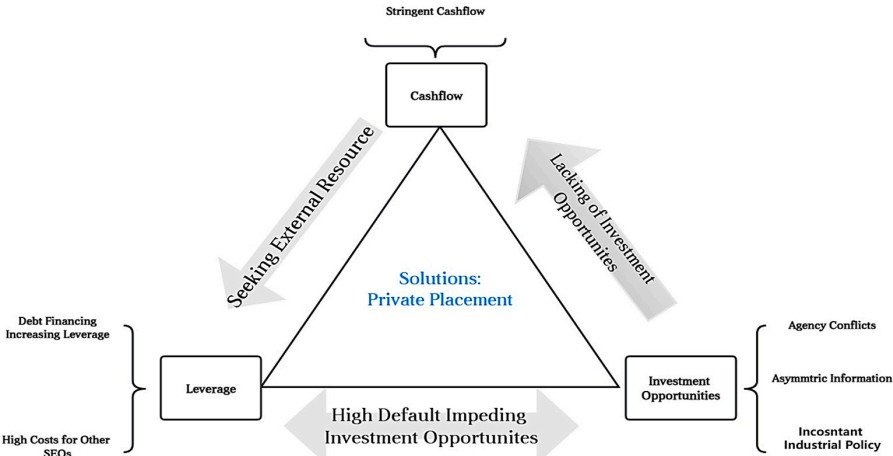

**Figure 4.** Private placements as a micro-level solution for real estate company risk management (source: author).

1. Reducing Debt Burden:

Private equity placements can increase the equity capital of real estate businesses, lowering their leverage. The private equity placement is essentially a capital structure change event, which reduces the debt ratio of the firm by increasing equity. Meanwhile, it is the freeing up of space for future financing (Bolton and Scharfstein 1990; Maksimovic and Titman 1991).

2. Supplementary Working Capital:

One of the most pressing issues faced by real estate companies is not the lack of assets but the shortage of working capital. The real estate industry inherently demands a considerable amount of transitional funds (Lam et al. 2011). Funds obtained through private placements can effectively mitigate the financial constraints in the company's cash flow.

3. Improved Corporate Governance:

Private placements can alter the ownership structure of a company by introducing external investors, thereby enhancing the balances within the firm (Wruck 1989). According to the monitoring hypothesis, external investor oversight has the potential to improve capital utilization efficiency. Furthermore, major shareholders in Chinese real estate companies typically hold relatively high equity stakes, making concerns about equity dilution and potential loss of control less pronounced. They actively participate in these transactions, as evidenced by data from CSMAR, which reveals that major shareholders and their related parties participated in a significant portion of transactions; meanwhile, there is only one case of managerial private placement.[7] Building upon the preceding analysis, the involvement of major shareholders may also provide stronger supervision for the management, offering support for the company's growth.

4. New Opportunities Brought by External Stakeholders:

The key to enhancing an enterprise's cash flow is to find profitable investment projects on the verge of market saturation. Through private placement, businesses can expand their chances of finding new opportunities by sharing resource information with other external experts. At the same time, the involvement of these outside experts enhances the accuracy of the information. Habib and Johnsen (2000) used the real estate industry as an example to illustrate this mechanism.[8]

5.    The Need for Information by Real Estate Companies:

It is intriguing to note that real estate companies will encounter two opposing information asymmetries when evaluating new opportunities. Habib and Johnsen (2000) considered private placement to be a method of information disclosure. The companies benefit from having insider knowledge of the business's internal affairs, but they also asymmetrically perceive the outside world. Despite perceiving demand through recent contracts, companies remain unaware of changes in aggregate demand and other external factors. Even with successful information gathering from customers, distortions arise from customer exaggerations and careless remarks, creating a skewed market truth. Information distortions persist even when utilizing experienced consulting firms. Consulting companies often receive higher compensation for providing biased reports to businesses. Those producing reports aligning with management expectations find it easier to secure payment and future service opportunities. Private placement investors, often of considerable scale, possess access to in-depth information and expert analysis. The discount reflects information exchange between insiders and outsiders, along with liquidity compensation. External investors are willing to buy newly issued shares at a slight discount when optimistic about a project, and vice versa. This discount signifies an equilibrium between the expectations of companies and external expertise. Understanding the subscription of outside investors allows the issuing company to grasp the prospect and value of its project. This is particularly valuable for real estate companies dealing with projects of low circulation, challenging valuation, and a prolonged liquidation cycle.

6.    Private Placement's Certification Role:

Sophisticated investors are optimistic about the chance to revive the business if suitable investors are willing to participate in the private placement to the market. Participant investors are thought to have better knowledge than other investors. According to the certification hypothesis, this would convey to bondholders that the company's prospects are improving by capitalizing on the market's optimism. Whether or not that signal proves to be accurate, it may still help a company's finances. Listed companies can send positive signals to the market through private placement.

7.    Market Segmentation:

From the perspective of market segmentation, private placement results from the classification of the market by investors at different levels. The development of token blockchain technology and the advent of REITs enable ordinary investors to overcome regional differences and capital constraints, creating unique real estate investment opportunities (Hoesli and Oikarinen 2012; Smith et al. 2019). Placing participants, on the other hand, have a superior grasp of the seller as experienced investors, allowing them to mitigate firm-specific risks. They can even negotiate directly with businesses to come up with financing solutions that benefit both parties. As a result, the rise of private placements is a market-driven systematic arrangement aimed at more sophisticated and experienced investors.

8.    Other Benefits and Features:

The real estate sector enjoys an inherent advantage in information accessibility. Given its close connection to people's lives, the general public can easily understand the industry's fundamentals through various media. It is absurd to expect an average person with basic education to know more about complex machinery than about local house price variations. This combination of predictable investment returns and reduced risk in rent revenue changes make it easier for the real estate business to secure refinancing opportunities. Furthermore, China's publicly traded enterprises have a clear preference for equity funding (Huang and Zhang 2001). These interests jointly determine that Chinese real estate enterprises regard private placement as an important refinancing means and risk management strategy.

## 8. Concerns on Private Placement

Private placements as a risk mitigation strategy for Chinese property companies give rise to several concerns that need to be noted. Firstly, existing research has predominantly focused on market-level analysis, with limited exploration in the context of the Chinese real estate industry. Questions arise about whether private placements in Chinese real estate companies exhibit the same patterns as other markets. Moreover, the motivations of participants may vary based on specific industries and transaction contexts. Secondly, the involvement of major shareholders in private placements within the real estate sector is widespread. It is imperative to elucidate the roles played by major shareholders when participating in private placement transactions. Evidence by X. Zhang (2017) suggested major shareholder participation yields more favorable announcement effects than institutional investors, possibly due to perceived informational advantages, implying a monitoring role. However, conflicts with other shareholders support the entrenchment hypothesis, warranting further research (Tong 2014). Thirdly, in line with agency theory, equity financing imposes fewer managerial constraints compared to debt financing. Despite the general notion of equity financing being costlier than debt, some scholars argue that the cost of private placements in China may be lower than perceived (Lu and Ye 2004), which can impact real estate companies' choice of financing channels. Fourth, it is important to consider the potential risk of reverse selection. In China's unique hybrid economic model, government intervention can lead to adjustments in real estate policies, as exemplified by the central government's accommodative policy in August 2023. However, such interventions must be carefully monitored to avoid adverse selection issues, where capital flows into low-yield industries at odds with the government's economic restructuring goals. Fifth, the introduction of REITs in China's securities market in 2021 is a significant development. Experience from Western markets suggests that private placements in the context of REITs may differ from traditional private placements, demanding further research into China's REITs practices. Finally, it should be noted that, when approving private placement deals, there should be no difference due to the different ownership attributes (Yang et al. 2016).

Overall, private placement as a risk-mitigation method should consider the transaction's monitoring and certification impacts to minimize entrenchment and tunnelling effects. Understanding the mutual operation of important stakeholders and uncovering the phenomena of private placement of Chinese real estate firms are beneficial to increasing real estate enterprise production efficiency and shifting the development mode of real estate enterprises.

## 9. Conclusions

Never have real estate and finance been so tightly intertwined. Land finance has played a pivotal role in China's economic miracle over the past decades, yet it has accumulated significant risks, including extensive bank exposure and high firm leverage. This review explores the analysis of private placement, positioning it as a market-oriented financing channel for Chinese real estate enterprises. Private placement not only provides an alternative to the traditional banking system but also contributes to reducing banks' risk exposure. Consequently, they assist the government in restructuring the economy and promoting the healthy development of the real estate sector.

As a risk mitigation strategy, private placements effectively lower the elevated leverage of real estate enterprises, easing financial constraints and mitigating information asymmetry. Real estate firms can gauge market prospects for projects through the transaction's discount rate while leveraging communication with sophisticated investors and experts to access additional investment opportunities. These measures are particularly beneficial for Chinese real estate enterprises currently grappling with debt and operational crises. However, our understanding of the intricate dynamics and internal mechanisms of private placements within Chinese real estate enterprises remains conspicuously inadequate.

Previous research on private placements has predominantly centered on the market level, with only a limited number of studies shedding light on the nuances of private

placements in the Chinese real estate sector. These findings have exhibited disparities when compared to earlier research and have also diverged from the experiences of other nations' real estate industries. This accentuates the urgent need for further research in this domain.

Due to the lack of sufficient literature, we cannot fully explain the issue of China's real estate private placement, but it also provides a direction for future research. Numerous intriguing questions beckon exploration. For instance, do market performance patterns fluctuate across different time intervals? Are there distinctive transaction characteristics? How do the various participants interact within this framework? Do major shareholders employ private placements to bolster corporate governance or do they exploit them to the detriment of other shareholders? What factors exert influence and how do they manifest in these transactions? The absence of answers to these critical questions leaves policymakers devoid of a solid theoretical foundation for the establishment of institutional rules and regulations. Investors are left without adequate guidance on how to effectively engage with real estate companies' development, and real estate enterprises find themselves adrift without clear direction on how to harness private placements to secure additional capital, optimize their capital structures, mitigate information asymmetry issues, and enhance overall management efficiency. Consequently, delving into the realm of private placements in Chinese real estate not only promises to enrich academic understanding but also holds the potential to offer enhanced guidance for the healthy and sustainable development of the real estate industry.

**Author Contributions:** Conceptualization, Y.N. and R.B.A.J.; Writing—original draft preparation, Y.N.; Writing—review and editing, Y.N. and R.B.A.J.; Supervision, R.B.A.J.; All authors have read and agreed to the published version of the manuscript.

**Funding:** This research received no external funding.

**Data Availability Statement:** The data used in this study can be found from China Stock Market and Accounting Research (CSMAR) database and Tonghuashun Wencai Database.

**Acknowledgments:** The authors express special gratitude to the anonymous reviewers for their valuable comments and guidance in the early versions of the manuscript.

**Conflicts of Interest:** The authors declare no conflict of interest.

## Notes

[1] The latest average price was calculated by dividing the total money traded by the total traded shares in the same period.

[2] Those types of investors are public listing companies' block shareholders, real controllers of the company, strategic investors or their arm-length connected persons, and the person or organization that has absolute control over the company through this private placement deal. Any individual or organizational investor not included in the list shown above must lock their newly bought shares for up to 12 months. Since February 2020, the lock-in period for regular private placement investors has been reduced from 12 months to 6 months, while the related party is locked from 36 months to 18 months.

[3] New amendments include "A Decision on Amending the Administrative Measures on The Issuance of Securities by Listed Companies","A Decision on Revising Interim Measures for The Administration of Securities Issuance of GEM Listed Companies", and "A Decision on Amending Detailed Rules for The Implementation of Non-Public Offering of Shares of Listed Companies" (from now on referred to as the New Rules on Refinancing).

[4] They used a meta-analysis method and revealed that the average cumulative excess return is −0.98 percent and the median is −1.39%, which is statistically significant. Non-private placement outlier returns, on the other hand, are less unfavorable.

[5] There is, of course, knowledge asymmetry among investors. Investors who get an informational advantage will continue to push out others who do not. However, similar information asymmetries occur even among the best-informed investors. The issuance market will finally fail after all investors have been eliminated from the market by this method of elimination. In order to attract or compensate the information-disadvantaged, the issuing business and underwriters employ a reduced offering to persuade investors to subscribe to the new shares (Rock 1986). Comparable mechanisms are also presented in the work of (Beatty and Ritter 1986).

[6] The data for the year 2023 is as of 24 November 2023.

[7] From 2006 to July 2023, major shareholders or their related parties participated in 87 of 144 real estate private placement cases. "ShiLianHang" is the only case involve in managerial private placement.

8    They present an example: "Local Commercial Real Estate Developers.... but they often lack knowledge about the local, regional, or national demand for office space. . ., real estate investment trusts (REITs) specialize in gathering this information. A local real estate developer and a REIT May,... ensure that the REIT provides a credible forecast of local office demand. The price . . . reflects its specialized expertise in estimating office demand and in valuing and operating office buildings. The REIT becomes a residual claimant to the accuracy of the forecast embedded in its equity claim, . . ..".

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
