# Peer review of "Private Placement of China-Listed Real Estate Firms: A Conceptual Idea"

_jrfm, doi:10.3390/jrfm16120516_

Round 1

Reviewer 1 Report

Comments and Suggestions for Authors

The paper reviews the theory and empirical evidence on the private placement around the world and is trying to related the findings in the previous literature to the private placement of Chinese real estate firms. 

While the literature review is complete, the authors made a very loose link with the private placement of the Chinese real estate firms. Thereby, it is somewhat  hard to understand how the theory and evidence developed in other countries can be applicable to the Chinese cases. Although the authors mention the unique aspects of the Chinese real estate business environment, there is no clear indication how the previous findings are related to the Chinese cases. When the main theme of the paper is the unique aspects of the private placement by Chinese real estate firms, the argument of the authors shall be more focused on the issues.  

I recommend the authors to conduct some empirical tests to provide solid theoretical and empirical evidence.  The langus

Comments on the Quality of English Language

The writing can be more concise.  For example, "The implementation, regulation, and practices in different nations are highly diverse 30 due to large variances in history, economic growth, financial markets, society, and legal systems" ...the authors tend to write this way to include everything in one sentence. 

Reviewer 2 Report

Comments and Suggestions for Authors

General assessment

The present paper mobilizes a set of ideas relating to the prior literature on private placement as an important refinancing and risk reduction strategy for Chinese listed real estate companies.

The major interest has to do with how critically it explores the comparison of placement techniques in various countries to comprehend the understanding of private placement in China-listed property companies.

It is well written and structured, and in spite of some theoretical, conceptual and methodological shortcomings, we recommend the article for publication, but only after some changes introduced by the author. The suggested changes, some more profound, others are more superficial, follow the next point with a more specific comment.

Specific assessment

1.      The introduction is very vague about the intentions and objectives of the paper. Despite the relevance of the key issues raised in the abstract, these introductory paragraphs can hardly be considered a good introduction. The author should therefore review the construction of the introduction. The introduction it must be an overview of the contents of the research in the paper without going into too much detail. Only a few paragraphs are enough. Briefly describe the importance of the study area. Specify the relevance of the publication of the current paper, ie, explain how this present work contributes to the progress of knowledge in this line of research in real estate studies. The author also needs to define better what PIPE and IPO are.

2.      The author needs to recognise in its paper the influence of the risks of financialisation of housing and real estates and land all over the world. Housing and real estate markets worldwide have been transformed by global capital markets and financial assets. Known as the financialization of housing, the phenomenon occurs when housing is treated as a commodity and a financial asset—a vehicle for wealth and investment—rather than a social good. With roots in the 2008 financial crisis, the impact of the shift from housing as a place to build a home to housing as an investment has been devastating. This includes millions of evictions as a result of foreclosures in countries most affected by the Global Economic Crisis.

Several facts featuring the financialisation of housing, rental and real estate markets: the link between the support of homeownership, mortgage debt and securitization, which has been one of the causes of the global financial crisis; the  liberalization  and financialization  of rental markets; the dismissal of public housing, financialization of housing companies, and recent emergence of financialized forms of social rented and affordable housing; recent reforms that have made real estate and housing markets  increasingly  attractive  to  financial  investments;  fast-growing  concentration  of  housing ownership, and penetration of institutional investors and specialized companies (including so-called ‘vulture funds’) through the purchase of massive housing stocks. The authors need to adress the negative consequences of financialisation of real estate for the stability of these markets and all productive economy.

3.      Lines 84-90: In this sense, the creation and strong regulation instituted by the China Securities Regulatory Commission is positive as it is to regulate the offering, trading, cross-border conversion, information disclosure, and other activities in relation to Depository Receipts Business under the stock connect scheme between domestic and overseas stock exchanges, to protect the legitimate rights and interests of investors, and to maintain the order of the securities markets.

4.      Lines 534-543: The authors say that “China's unique political structure sets it apart from developed nations' markets, which are dominated by private corporations. State-owned or collective enterprises must be managed by specific individuals because a representative needs to be selected to exercise control and management power. As a result, these businesses are inevitably vulnerable to agent issues. Managers are motivated to work for their interests even though they lack the incentive to maximise the interests of shareholders (i.e., all citizens or the collective). It stands to reason that private companies will profit more than public ones.” In this paragraph the authors make a moral, contradictory and even prejudiced judgment, which is why they must rethink it because as it is does not correspond to the truth in general. In China, the economic system is a unique blend of socialism and market-oriented reforms, commonly referred to as "Socialism with Chinese Characteristics." Some of the key concepts of socialism in China are: a) State ownership, while there has been a significant amount of privatization in recent years, many of China's largest and most influential companies are still owned and controlled by the state; b) Central planning, The government sets targets for economic growth and makes strategic investments in industries and infrastructure to achieve those targets; c) Market-oriented reforms, China has embraced market-oriented reforms, allowing for private businesses, foreign investment, and entrepreneurship. However, the state still controls the economy, particularly in strategic industries. Communist Party leadership: The Chinese Communist Party maintains a firm grip on the country's political and economic system, with many state-owned enterprises and key industries being led by Communist Party members. In this sense, state-owned enterprises and key national strategic industries must be managed by administrators and members of the government, as they are positions of great economic and social responsibility, being completely trustworthy. Overall, China's economic system represents a unique combination of socialist elements, with significant government control and economic intervention. While this has led to impressive economic growth and development, making the Chinese financial system resilient and less susceptible to threats from the instability of the global financial system, which is inherently speculative and does not work towards progressive purposes towards a productive economy and social development. For instance China's private equity and venture capital (China PE/VC) market is large and growing, and currently represents the second largest PE/VC market in the world. We believe an allocation to China PE/VC can add value within a broader global PE/VC portfolio by enhancing returns and diversification. Now, is important for the authors to understand that public investment can, again directly or indirectly, create favorable conditions for private investment, for instance, by providing infrastructure such as roads, highways, sewage systems, and harbors. Better facilities may increase the productivity of private investment and reduce the cost of production of the private sector, a positive impact on the profitability of private investment. This would result in a “crowding-in” effect on private investment. Furthermore, government spending itself may directly crowd in private investment, by contracting directly with private. State enterprises can also subcontract to private firms, directly increasing private investment (See Xu and Yan, 2014 here: https://www.tandfonline.com/doi/pdf/10.1080/17487870.2013.866897

5.      When the authors advocate that “Managers are motivated to work for their interests even though they lack the incentive to maximise the interests of shareholders (i.e., all citizens or the collective)” they forget to recognize that it is capitalism that emphasizes private property rights, with individuals having the right to own and control property. In contrast, socialism emphasizes collective ownership and the community's interests over individual interests. For example, in a socialist economy, land and natural resources may be collectively owned by the state rather than individuals owning and controlling those resources. About distribution of income: Capitalism is frequently linked to income inequality, with money accumulated in the hands of a small number of people and businesses. Socialism, in contrast, promotes income equality and transfers money from wealthy individuals to the poor. It´s capitalism that strongly emphasizes individual choice and self-interest, initiatives that promote social welfare are frequently seen as a drag on the economy. Socialism, on the other hand, emphasizes social welfare more and calls for the government to provide all citizens with access to essential services like healthcare and education.

6.      In conclusions, maybe add that China’s economic planning is creating a new department to help private businesses, the latest step by the government to revive confidence in the sector and bolster growth. The private economy development bureau will be responsible for tracking and analyzing the state of the industry, along with coordinating and drafting policies to promote its growth, the National Development and Reform Commission announced in the beginning of September 2023. Beijing has unveiled a drip-feed of policies in recent months intended to revitalize private companies and attract foreign investment, vowing to ease market access and create opportunities for global cooperation. It is rare for the government to set up an agency specialized in a certain sector. It sends a policy signal for guiding expectations in an institutionalized way. The new bureau will also regularly talk to companies and help them resolve their main problems, as well as support their attempts to improve international competitiveness. In last July, the ruling Communist Party and the government pledged to treat private companies the same as state-owned enterprises — a move seen by investors at the time as a framework for future support.

7.      We can also find some limitations in the paper in terms of references, including the lack of standardization and compliance, in its final list. The section of the bibliography in its implementation requires time and detailed attention to certain details. The author must make an effort to maintain the required style and standards of publication in the journal for all references cited and not just some.

Final assessment

The paper presents an important contribution to risk and finance studies, presenting a good empirical demonstration of results and being drafted clearly and objectively. Also reveals a good scientific terminology and vocabulary in the study of finance instruments in real estate market.

Despite the requested changes, the author should understand that his/her paper is very good and just sometimes what it takes is to add a small paragraph or phrase on the subjects required, or even change or replace a reference, to significantly improve the work. In short, in spite of the considerations outlined in the field of specific assessment, this review is in favor of publishing the manuscript but with minor modifications, with the expectation that this will surely be an important contribution to the academic and political debates surrounding the investment in real estate markets.

Comments on the Quality of English Language

 Minor editing of English language required

Reviewer 3 Report

Comments and Suggestions for Authors

The paper deals with an interesting research topic. The review examines Private Placement transactions, one of how new securities are issued on the primary market. Specifically, the authors analyse the private placement of listed real estate companies in China. Through this capital-raising mechanism, real estate companies select private or institutional investors instead of offering their securities to the general public. This approach allows companies to raise funds more selectively. Subsequently, the risks of private placement are analysed. The work is interesting and shows a good understanding of the topic. However, the general structure of the paper remains to be improved. Above all, the abstract, introduction and conclusions need to be improved. These sections should be interconnected, in the sense that the conclusions must contain the answers to the research questions introduced in the introduction (questions that are currently missing).

Below I report some considerations aimed at better highlighting aspects that are already partly present in the work:

1)   Main issues.

Abstract

The abstract lacks some important aspects that could justify the need for a review: what are the gaps or current problems that arise from the analysis of the reference literature on the topic? What are the indications and suggestions you give to scholars at the end of your analysis?

In addition, the methodological approach you have adopted for the selection of the references and documents analyzed should be briefly described. Finally, the main conclusions you have reached should be summarised.

Introduction

Although a brief introduction is suitable for a review, it lacks many elements useful to understand the importance of the work. You should focus on the purpose of the review. What is the reason that prompted you to analyze this topic? Even for a review, the purpose of the work should emerge from the introduction. Why is it relevant to the scientific community? What are the gaps that emerge from the literature analysis, then proposed in the following sections? What is the connection with the issue of risk? In summary, what are the types of risks of this investment policy? All these aspects are missing in the introduction, which should give the reader a general and concise overview of the study. In the introductory section, you should explain all the points you will develop in the following sections. At the moment, there is only the definition of a private placement.

Summary

Some research questions are present in the Summary (lines 850-852), but they should already be presented in the introduction (and further explored). In your conclusions, you could briefly describe the practical benefits for stakeholders that can be derived from the study. You affirm that further studies are needed in the future. But you don't give any clear guidance for researchers. Based on your analysis, what specific aspects deserve to be explored in future research? What doubts are still open? In the conclusions, before suggesting further future investigations, summarize the limitations of the study.

2)  Further issues.

·   Why is "Table 1" bold in the text? (see line 32). It does not seem to me that this respects the format of the magazine.

·   In Table 1, the name of the country is Australia and not Australian (the latter is an adjective).

· Also in Table 1 I would include at least one other country representative of Europe and the American continent, to have more than one example per continent.

· In section 2.1. the regulations of all Table 1 countries except Sweden are explained. I'd say add a summary of the Swedish regulations (and any other countries you might add) as well.

· Figure 1 (see line 91) should also not be bold. Ditto for all the in-text citations in the following figures and tables.

·  In the caption of Table 1 the sources should be reported (or specify that the image is your processing/reworking).

· In section 4.1. You should include a summary table of the literature consulted for each of the three categories of private placement theories. For each category you could indicate the author and year, name of the article, Journal/Book title, Publisher, and summary of the argument covered.

· In Section 7, before looking at the risk to real estate companies, I suggest that we first look at the risk in the real estate and construction sectors as a whole and in general terms (adopting an international perspective). In this sense, the reasons that can make real estate and construction investments financially sustainable or partly risky could be investigated. The following references may be useful for this:

Derisking Real Estate in China’s Hybrid Economy (No. w31118). National Bureau of Economic Research

Tax Policies for Housing Energy Efficiency in Italy: A Risk Analysis Model for Energy Service Companies. Building 13(3). 2023.

Identifying principal risk factors in Turkish construction sector according to their probability of occurrences: a relative importance index (RII) and exploratory factor analysis (EFA) approach. International Journal of Construction Management, 23(6). 2023.

· See lines 682-684: The way the sentence is constructed means a forecast of value by 2020. But 2020 is already over. The sentence should be reworded in the past tense. And then, do you know if the prediction has been confirmed?

· Instead of a summary, I would call the last section conclusions.

· Prepare the bibliography according to the format of the journal.

Comments on the Quality of English Language

Minor English language revision required. Some suggestions are in the previous comments.

Reviewer 4 Report

Comments and Suggestions for Authors

Review for “Private Placement of China-Listed Real Estate Firms: A Conceptual Idea”

This paper reviews how private placement is being conducted in the Chinese real estate industry.  The review is comprehensive, but I have several questions: Why is it important to investigate the real estate setting in China for private placement? Is there any evidence of the current private placement size for the Chinese real estate market? Note Chinese firms have regulatory constraints in SEO, so how does that affect the popularity of private placement? Another example is whether the state-owned structure is more prominent for real estate than the rest of the Chinese firms. In summary, the issue with many parts of the review is that it serves more as a general review of private placement literature, without adequate justification on why this paper is about the Chinese real estate industry. While we know the real estate sector is important for the Chinese economy, there seems to be a missing link for the importance of private placement for the Chinese real estate industry. While acknowledging the review contains sections discussing the development of the Chinese real estate industry and risks of these companies, how are these companies linked with the enhanced importance of private placement? Do we have some stylised facts of how financing has been conducted for Chinese real estate firms such that these firms are unique and different from other Chinese firms?

Several minor issues: The introduction could be lengthened and strengthened. For instance, it should cover the motivation of the current review, and summarize the findings. There could also be some rearranging of the sections, as some sections are China-specific and some are about general literature, and this goes back and forth.

Round 2

Reviewer 1 Report

Comments and Suggestions for Authors

The  writing  has  improved  from the  previous  version. Nonetheless it  still lack  some  data  to  provide  solid  reasoning.  I  understand  the  purpose  of  the  paper is  more  about  the  theory. However  the authors  could  provide some  statistics  that  can  support  their  argument..

Comments on the Quality of English Language

English  is  fine.

Reviewer 3 Report

Comments and Suggestions for Authors

The authors responded to all comments and made necessary changes to the paper following the previous review cycle. It can now be accepted for publication. I suggest paying attention to the formatting of the text (lines 84-94 are in bold). Re-read the work carefully before final submission.

Reviewer 4 Report

Comments and Suggestions for Authors

No further suggestion.

Round 3

Reviewer 1 Report

Comments and Suggestions for Authors

It appears that the authors made additional works to improve the paper.  

Comments on the Quality of English Language

English is fine for publication.